# The Effect of Quinoa Seed (*Chenopodium quinoa* Willd.) Extract on the Performance, Carcass Characteristics, and Meat Quality in Japanese Quails (*Coturnix coturnix japonica*)

**DOI:** 10.3390/ani12141851

**Published:** 2022-07-21

**Authors:** Shaistah Naımatı, Sibel Canoğulları Doğan, Muhammad Umair Asghar, Martyna Wilk, Mariusz Korczyński

**Affiliations:** 1Department of Animal Production and Technologies, Faculty of Agricultural Sciences and Technologies, Niğde Ömer Halisdemir University, Niğde 51240, Turkey; shaistah_naimati@mail.ohu.edu.tr (S.N.); scanogullari@ohu.edu.tr (S.C.D.); 2Department of Animal Nutrition and Feed Sciences, Wroclaw University of Environmental and Life Sciences, 25 C.K. Norwida St., 51-630 Wrocław, Poland; martyna.wilk@upwr.edu.pl (M.W.); mariusz.korczynski@upwr.edu.pl (M.K.)

**Keywords:** quail, meat quality, shelf life of meat, quinoa seed, performance

## Abstract

**Simple Summary:**

There has been a rise of interest in using natural herbs as antibiotic alternatives or natural feed additives in diets to enhance animal productivity and maximize potential production during the last decades. Quinoa seed extract (QSE), which has a high antioxidant activity and phenolic content, is one of the natural feed additives. Quinoa (*Chenopodium quinoa* Willd.), a gluten-free pseudocereal, has grown in popularity over the years. Quinoa is a good source of protein (vital amino acids like lysine and methionine), carbohydrates, fiber, tocopherols (vitamin E), unsaturated fatty acids, and polyphenols. This research aimed to evaluate the effects of different amounts of QSE in the Japanese quail diet on growth, slaughter carcass, sensory characteristics, and certain meat preservation capabilities. The addition of QSE had a good effect on quail weight gain and growth of animals, lipid profile, antioxidant, immunity, meat storage quality, pH, and pathogenic bacteria content, according to our findings. It is worth mentioning that QSE reduced overall bacteria levels while improving meat preservation quality. According to the presented research, the best results of quail performance were obtained with 0.2 g/kg and 0.4 g QSE/kg of the quail’s fodder. While the addition of 0.4 g QSE/kg of the quail’s fodder had a significant effect on meat shelf life and could be used in poultry mixed feed to prevent or delay lipid oxidation of meat.

**Abstract:**

This research was conducted to determine the effect of quinoa seed (*Chenopodium quinoa* Willd.) extract on the performance, carcass parameters, and meat quality in Japanese quails. In this study, 400 quail chicks were divided into a control group (without quinoa seed extract addition) and 3 experiment groups (4 replicates containing 25 quails in each). Commercial feed and the addition of different concentrations of quinoa seed extract (QSE) 0.1 g/kg, 0.2 g/kg, and 0.4 g/kg were used in the study. During the second week of the experiment, the highest feed intake was obtained from the supplemented groups (*p* < 0.01). After 5 weeks of experimentation, the highest feed consumption was noticed in the group with 0.4 g of QSE additive. The QSE additive affected the live weight gain values of all experimental groups during 1 week of the experiment. The highest values of hot carcass weight were noticed in groups with 0.2 and 0.4 g of QSE additive (*p* < 0.01). While the highest value of cold carcass weight was noticed in a group with 0.2 g of QSE additive (*p* < 0.05). Thigh, breast, back and neck ratio, and internal organs (except gizzard) were not affected by the supplementation of QSE. As a result of storage of breast meat at 4 °C for 0, 1 days, 3 days, 5 days, and 7 days, it was determined that the number of pH, thiobarbituric acid, peroxide, and total psychrophilic bacteria were lower in the groups with QSE as compared to the control group (*p* < 0.05). In conclusion, the best results of quail performance were obtained with 0.2 g/kg and 0.4 g QSE/kg of the quail’s fodder. While the addition of 0.4 g QSE/kg of the quail’s fodder had a significant effect on meat shelf life and could be used in poultry mixed feed to prevent or delay lipid oxidation of meat.

## 1. Introduction

Quails inhabit regions of Asia, America, Europe, and Australia; however, remunerative quail breeds are raised for eggs and meat intention around the globe. Japanese quails (*Coturnix coturnix japonica*) can attain a live weight of 200 g within 4 weeks of age. In a free-range rearing system, the weight of this bird is around 100–160 g [1]. Regardless of the quality, quail farming is very expedient due to the least cost of maintenance, healthy production, and a remarkable revenue ratio [2]. The different quail products have shown numerous pro-health properties, helpful in the treatment of different diseases like ulcers or gastritis, consolidation of heart muscles, and rehabilitation of blood circulation after blood stroke, and had antineoplastic effects [3]. In recent years, the trend of healthy, natural, and safe food consumption has been increasing, depending on the food and health concerns of people. In addition, people are showing more interest in foods containing bioactive or functional components that will provide additional benefits to their health [4]. This situation depicted less use of synthetic additives and the production of less processed food products without sacrificing food safety for the production of food products in line with consumer demands. In general, the consumer perception towards the purchase of meat and meat products is that it is unhealthy because those foods are thought to increase the risk of cardiovascular diseases, obesity, and cancer due to their high-fat content [5]. Various synthetic and natural antioxidants have been used in many countries to prevent undesirable reactions and extend the shelf life of the product. However, recent trends force the use of natural preservatives instead of synthetic ones due to the carcinogenic or mutagenic potential caused by high-dose intake [6]. Antibiotic remains found in meat and eggs may cause complications in human health. Phytobiotics are utilized to enhance growth and improve carcass quality in broiler meat with lower fat content [7,8,9]. Poultry meat, rich in unsaturated fatty acids, has increased susceptibility to oxidative degradation [10]. The most serious issues encountered in meat processing, cooking, and cold storage include the oxidation of meat lipids. Consequently, it reduces meat shelf life, degrades food flavor, and affects organoleptic qualities. The use of natural antioxidants for increased oxidative stability of meat is a topic of great interest today. Phenolic compounds (PCs) have long been recognized for their nutritional and therapeutic advantages, which include antioxidant and antibacterial characteristics. While a wide variety of natural products, such as antioxidants, have shown promising results, many of the sources used may contain toxic and antinutritional factors and can have adverse effects when used in large quantities [4,11]. Natural and synthetic antioxidants that prevent lipid oxidation should be supplemented in the diet to sustain the quality of food products [12,13]. Therefore, further studies are needed to characterize the active compounds in these natural sources and evaluate their efficacy, safety, and stability in different amounts and different products.

Following the restriction of using antibiotics as a growth regulator in the poultry industry, the quest for natural and safe compounds has started that have no harmful effects on human and animal health subsequently. At this point, researchers have been focusing on medicinal and aromatic plants. One of the plants studied in recent years is the quinoa (*Chenopodium quinoa* Willd.), a pseudocereal crop, traditionally cultivated for 5000 years in the Andean region of South America and basically used in the same way as wheat and rice. Although quinoa is a less well-known plant, interest is increasing due to its extraordinary nutritional value and health benefits compared to other grains. Quinoa is suitable to be grown in all regions from sea level to high altitudes. In addition, it has a wide genetic diversity that can adapt to salinity, drought, and cold regions [14,15]. Quinoa is a pseudocereal belonging to the Chenopodiaceae family, has been accepted as a functional food, and has gained increasing importance in recent years due to its high nutritional value. The quinoa plant and seed are not only rich in macronutrients such as protein, polysaccharides, and fats but also micronutrients such as polyphenols, vitamins, and minerals [16]. Quinoa is an important source of antioxidants, which can delay or inhibit the oxidation of lipids or other molecules by preventing the initiation or propagation of oxidation chain reactions [17]. These functional properties can make a strong contribution to human nutrition, especially in protecting cell membranes [18].

## 2. Materials and Methods

Japanese quail (*Coturnix coturnix japonica*) were used as animal material in the study due to its smallest size and fast-growing rate. A total of 400 quails (day old, DOC) were obtained from breeding quails in Çukurova University, Faculty of Agriculture. The experiment with animals was carried out at Niğde Ömer Halisdemir University, Ayhan Şahenk Agricultural Research Application and Research Center. Laboratory studies were carried out in the Niğde Ömer Halisdemir University, Faculty of Agricultural Sciences and Technologies, Department of Animal Production and Technologies, Turkey.

### 2.1. Arrangement of Experiments

The experiment was continued for 35 days; during this period, feed and water were given ad-libitum. Commercial broiler chick starter feed containing 23% crude protein and 3100 kcal/kg metabolic energy was used in the feeding of quails. Feed was given to quails for 5 weeks, according to the nutritional requirements of quails [19]. The study consisted of 4 groups (4 replications with 25 chicks in each group) containing 0 g/kg, 0.1 g/kg, 0.2 g/kg, and 0.4 g/kg quinoa seed extract. Natural and artificial lighting was applied for 24 h. With the air conditioner in the quail rearing room, the room temperature was adjusted to a suitable level and controlled with a thermometer. The temperature was adjusted to 32–33 °C in the first week with the thermostatic heater in the main machines, where the quails were placed, and in the following weeks, the temperature was lowered by 2–3 °C every week and fixed at 24–25 °C.

### 2.2. Quinoa Seed Extraction Process

Quinoa seed extract was obtained from white quinoa seed grown in the Niğde region. To obtain the extract, the quinoa seeds were washed and left to dry for 24 h in a drying cabinet set at 60 °C. Thoroughly dried quinoa seeds were ground and subjected to extraction with different ethanol and water levels (70% ethanol: 30% water and 80% water: 20% ethanol). A sample of 10 g of ground quinoa seeds was taken for each extraction process. Then, the quinoa was dissolved by keeping it on an orbital shaker (500 RPM, 24 h, 40 °C; Boeco, OS20, Hamburg, Germany). The dissolved mixture was then filtered with coarse filter paper and then ethanol was evaporated at 50 °C in a rotary evaporator to obtain quinoa seed extract [20,21]. The obtained extract was stored at −80 °C to determine the total phenolic compound and antioxidant content. The reason for using different ratios of solvents used in extraction is to determine the phenolic and antioxidant substance content in both processes and to use the extract with higher content in quail compound feed in the research.

### 2.3. Determination of Total Phenolic Content and Antioxidant Capacity of Quinoa Seed Extract

The total phenolic content (TPC) and antioxidant activity of QSE were examined. TPC was assessed colorimetrically using the Folin–Ciocalteu reagent (FC) as modified by Chuah et al. [22]. After 5 min, a 0.5 mL aliquot of the quinoa extract solution is transferred to a glass tube, and 2 mL of Na_2_CO_3_ solution (200 mg/mL) is added and agitated. The sample was then combined on a homogenizer, and the reaction was allowed to run for 15 min at room temperature. Ultra-pure water (10 mL) was added and centrifuged for 5 min at 4000 rpm to separate the sediment. Finally, the absorbance of the samples was taken at 725 nm in a spectrophotometer (Spectronic^®^ 20 Genesys M131, Illinois, USA) and contrasted to a gallic acid (Merck KGaA, Darmstadt, Germany) calibration curve. The results were given in mg gallic acid/100 g dry matter. All parameters were measured three times. Trolox Equivalent Antioxidant Capacity (TEAC) analysis is an analysis based on the inhibition of the absorbance of the 2,2’-azinobis 3-ethyl-benzothiazole 6 sulfonate (ABTS) radical cation by antioxidants [23]. To determine the antioxidant activity in quinoa seed extract, ABTS solution was first prepared. For this, a 7 mM ABTS solution containing 2.45 mM potassium persulfate (K_2_S_2_O_8_) was prepared and left in the dark for 12–16 h at room temperature to obtain a radical solution (ABTS+•). Since the antioxidant activity of quinoa seed extract is Trolox, a series of concentrations of both the extract and Trolox were prepared. The sample (10 µL) was added to 1 mL of ABTS+• and a decrease in absorbance was observed in spectrometry for 6 min. The slope was calculated from the graphs plotting the concentrations versus percent inhibition. As a result of the ratio of the slope of the quinoa extract to the slope of the Trolox concentrations, the antioxidant activity of the investigated antioxidant as 1 mM Trolox was determined [24]. While determining the antioxidant activity, three parallels were made for each concentration level and the measurements in the spectrometer were determined with micro cuvettes at 30 °C.
$$\mathrm{TEAC}\;\mathrm{value}\;\mathrm{µM}\;\mathrm{Trolox}\;=\frac{slope\;of\;the\;sample}{slope\;of\;the\;trolox}\times dilution\;factor$$

### 2.4. Determination of Live Weight Gain of Quails

The weekly live weight gain of quails was computed by subtracting the previous week’s average live weight from the average live weight of each replication in each group, indicating the weights for each week. During the research, the live weights of the quails were determined by weighting individually with an electronic scale with a precision of ±0.01 g, based on the day the research started every week. To determine the weekly live weight gains of quails, it was calculated by subtracting the average live weight of the previous week from the average live weight of each replication in each group determined in the weighting for each week.

### 2.5. Determination of Growth Performance and Carcass Characteristics

At the end of the trial (after 5 weeks), the average live weight value of each group (female and male separately) was determined. Quails were starved for 12 h before the routine slaughtering was performed. The quail’s wings, representing the male and female body weight averages of each group, were taken from each repetition. The feet of the quails were removed, and the warm carcass weight was determined after the internal organs were removed. The breast meat samples from each group were taken from the quails slaughtered and their carcass characteristics were determined. After the weights of internal organs such as heart, liver, and gizzard were taken, the carcasses were kept at 4 °C for 24 h, and at the end of this period, the cold carcass weight and the amount of abdominal fat were determined. The ratio of visceral weights to the carcass weight and abdominal fat weight to the carcass weight were determined. Carcass yield was calculated by considering the live weight and cold carcass weight at slaughter. The weights and lengths of the digestive system parts (esophagus along with the crop, heart, liver, gizzard, and abdominal fat) were determined in quails. While determining the length of the digestive system, its contents were not emptied, but when determining the weight, its contents were completely emptied, and its weight was taken. After the cold carcass weight was determined, to determine the proportions of the main carcass parts in the carcass, the weights of the main parts of the carcass (neck, back, thighs, breast, wings) were weighed with a scale with an accuracy of ±0.01 g, and their weights were recorded. The ratio of each carcass piece was calculated by proportioning the main carcass weights to the carcass weight.

### 2.6. Oxidation Analysis

This oxidation analysis included two parameters: peroxide value analysis and thiobarbituric acid number (TBARS).

To assess the oxidation state in the flesh from the samples from the quails maintained at 4 °C for 0 days, 3 days, 5 days, and 7 days, peroxide content analysis was conducted using the AOAC 965.33 technique [25]. Peroxide value analysis was performed to determine the oxidation state in the meat [26]. For this purpose, the breast meat was passed through a blender, and after homogenization, it was extracted, and oil was obtained. One ml of the oil obtained as a result of the extraction was taken and poured into 250 mL flasks and 30 mL of chloroform-acetic acid solution was added to it. Then, 1 mL of saturated potassium iodide solution was added, mixing well, and it was kept in the dark for 5 min. Then, 30 mL of distilled water and 4 drops of the starch solution were added and titration was performed with sodium thiosulfate solution. The titration process was continued until it became a light color and the amount of sodium thiosulfate spent in the titration was recorded [25]. The lipid oxidation status of two breast meat samples collected from each group stored at 4 °C after 0 days, 3 days, 5 days, and 7 days of preservation was determined by the thiobarbituric acid number analysis (TBA). A breast fat sample (0.1 g) was taken and transferred to a 25 mL balloon jug and made up to 25 mL by adding Butyl Hydroxy Toluene (BHT) solution to it. The mixture was then homogenized by mixing with the help of an ultra-thorax homogenizer. The homogeneous mixture was then poured into the beaker, 5 mL was transferred to the tubes, and 5 mL of thiobarbituric acid was added into the tube and mixed with the mixer again. The mixtures in the obtained tube were kept in a boiling water bath at 95 °C for 2 h. At the end of the period, the samples were read in a spectrophotometer at a wavelength of 530 nm [27].

### 2.7. Microbiological Analysis

Total psychrophilic viable counts were made at the end of 0 days, 1 day, 3 days, 5 days, and 7 days of storage at 4 °C of breast meat samples taken from quails. For this purpose, 10 g of breast meat samples were taken and mixed with 90 mL of 0.1% peptone water and homogenized for 1 min in the stomach. Serial dilutions were prepared by diluting the homogenized mixture with 0.1% peptone water and ¼ ringer solution. Plate count agar (PCA) medium was used in total psychrophilic viable count analyses. To obtain this medium, the dehydrated medium was dissolved in distilled water to 22.5 g/L. This medium was then sterilized in an autoclave at 121 °C for 25 min with all materials to be used. After this process, the medium was poured into 12.5 mL Petri dishes. According to the target number, sowing was done by pouring the homogenizer. The total number of psychrophilic bacteria was determined by the pour plate method. Petri dishes were incubated at 10 °C for 7 days. The number of microbial bacteria is expressed as log cfu/g [28,29].

### 2.8. Determination of pH in Meat

At the end of the experiment, three samples of breast flesh were collected from each subgroup, and the pH level of the breast flesh was evaluated at 1 day, 3 days, 5 days, and 7 days consecutively. On day 1, a Testo 205 meat and food pH meter was used to detect the pH level. Measurements were taken from three separate regions of the breast flesh for this goal, and the mean of these results was computed. The breast flesh was processed through a blender and combined with purified water after taking a 5 g sample and homogenized to measure the pH of the meat. The resulting homogeneous liquid was filtered, and the pH of the breast flesh was determined using a pH meter with a probe [30].

### 2.9. Statistical Analysis

Statistical Package for the Social Sciences (SPSS) 18 package program was used for the statistical evaluation of all data obtained at the end of the experiment by applying the variance analysis method. Significant differences between the groups were confirmed by Duncan’s multiple range test. Differences with *p* < 0.05 were considered as significant (a, b) and *p* < 0.01 as highly significant (A, B).

## 3. Results

### 3.1. Total Phenolic Content and Antioxidant Activity

To obtain quinoa seed extract, two types of extracts were prepared by using two different solvents (70% ethanol:30% water and 80% water:20% ethanol) and the total phenolic content was determined in each extract. While using 70% ethanol and 30% water in the extraction process, the total phenolic content of the quinoa seed extract was 1295.77 mg gallic acid equivalent per g, while using 80% water and 20% ethanol was only 287.01 mg GAE/g. Due to the fact that the total phenolic content was higher in the ethanol extraction, the extracts used in the research were obtained in this way.

The antioxidant capacity of the ethanolic extract of quinoa seed was obtained as 93.92 µmol Trolox/g.

### 3.2. Live Weight

At the beginning of the experiment, the average body weight in each group was quite similar and the body weight values in the groups varied between 8.42 g and 8.46 g. The results of weekly live weights of quails fed with mixed feeds containing different levels of quinoa seed extract (0 g/kg, 0.1 g/kg, 0.2 g/kg, and 0.4 g/kg) during the 5-week trial are given in Table 1. While there was no significant difference in body weight values between the groups in the 2nd, 3rd, and 4th weeks of the study, there were significant differences between the groups in the 1st and 5th weeks of the study (*p* < 0.01). During all weeks of the study, the groups supplemented with quinoa seed extract had higher body weight values than the control group. In the 5th week of the study, the highest body weight was obtained from the 0.2 g/kg quinoa seed extract added group (*p* < 0.01).

### 3.3. Live Weight Gain

Live weight gain values in individual experimental groups are given in Table 2 and Table 3. When examined, it was determined that there were statistically significant differences between the groups in terms of body weight gain only in the first week of the study (*p* < 0.01). The highest body weight gain was obtained from the group containing 0.2 g/kg of quinoa seed extract. There was no statistical difference in terms of body weight gain in all other weeks.

### 3.4. Feed Consumption

The effect of different levels of quinoa seed extract on weekly feed consumption was investigated (Table 4 and Table 5). It was observed that the feed consumption in the control group in all weeks during the experiment was lower compared to the groups with QSE. Statistically significant differences between the experimental groups were determined in the 2nd week of the study. The groups supplemented with quinoa seed extract consumed more feed than the control group (*p* < 0.01). Moreover, in the 5th week of the study, statistically significant differences between the experimental groups were determined (*p* < 0.05). The lowest feed consumption throughout the rearing period had the control group (*p* < 0.01; Table 5).

### 3.5. Carcass Values

There was no statistically significant difference between the groups in terms of the proportions of the live weight and carcass yield (Table 6). Quails from the 0.2 g/kg QSE and 0.4 g/kg QSE groups had significantly higher values of hot carcass weight (*p* < 0.01). The highest value of cold carcass weight had quails from the 0.2 g/kg QSE group (*p* < 0.05).

### 3.6. Carcass Part Ratios

There were no statistically significant differences between the groups in terms of the proportions of the main parts of the thigh, breast, back, and neck in the carcass (Table 7). The significant differences between the groups in terms of wing ratios were determined. The highest value of wings ratio was obtained in the group with 0.2 g/kg quinoa seed extract.

There were no statistically significant differences between the groups in terms of the proportions of the main parts of the heart, liver, and abdominal fat (Table 8). The significant differences in the weight of the gizzard were determined between the groups (*p* < 0.05).

The significant differences (*p* < 0.01) in the peroxide value of breast meat and the thiobarbituric acid value of breast meat during 1, 3, 5, and 7 days of storage between groups were determined (Table 9 and Table 10). The lowest values for these parameters were recorded in the group with the addition of QSE at the level of 0.4 g/kg.

At zero and 1st day, the total number of psychrophilic bacteria in breast meat was significantly lower (*p* < 0.01) in the 0.4 g/kg QSE group compared to others. On the 5th and 7th days, the total number of psychrophilic bacteria in breast meat was lower in the group with the addition of QSE at the level of 0.2 and 0.4 g/kg compared to other groups (Table 11). There were no significant differences between groups on the 3rd day.

### 3.7. Determination of pH in Meat

When the pH value of breast meat was measured (0, 1, 3, 5, and 7), quinoa seed extract-doped groups were found to have a lower pH value than the control group (*p* < 0.05). With the increase in the amount of quinoa seed extract in quail mixed feed, the pH value decreased (Table 12).

## 4. Discussion

Several PCs present in oil seeds have received a lot of interest due to their health effects, sensory-nutritional properties in food items, as well as their industrial applications. Alvarez-Jubete et al. [31] reported that quinoa seed extract had the highest phenol content (89.73 ± 1.74) and antioxidant activity (1586 ± 41.42) in water extraction. The quantity of phenolics in a product varies depending on the process of extraction. Bhaduri [32] reported that the quinoa seed extracts obtained from water, methanol, and ethanol showed significant antioxidant and phytochemical activities. In our research, the total phenolic content was higher in ethanol extraction compared to water extraction, which is why the extracts used in the research were obtained in this way. The difference in TPC and antioxidant activity between this study and earlier literature might be ascribed to QS cultivar changes in genotype, farming practices, and growing seasons.

This research was carried out to reveal the effects of the use of quinoa seed, which has high antioxidant activity and phenolic substance content, as a natural additive in quail compound feeds on performance and carcass characteristics, as well as the changes in quality during meat storage. Pompeu et al. [33] reported that the lectin found in quinoa seed has antimicrobial potential, which may explain a decrease in the total number of bacteria in breast meat with an increase in the QSE additive concentration (Table 11). Also, Marino et al. [34] investigated the effects of quinoa seed on meat quality and productivity of merino lambs. They reported that the addition of quinoa seeds to merino lambs feed enables better meat quality due to the close relationship between stress responses and the immune system.

The current study provides data on the impact of using different concentrations of QSE supplementation on quail growth performance, carcass features, lipid peroxidation during storage, microbial load, TBA values, and pH. The results revealed that adding QSE to the quail diets had no significant influence on weight gain during the 2nd, 3rd, and 4th weeks. Moreover, QSE treatment increased weight gain significantly in the first and fifth weeks, increasing feed consumption. According to the findings of this study, the inclusion of QSE enhanced quail feed intake. Amiri et al. [35] and Eassaway et al. [36] reported that feeding broilers a diet supplemented with QSE increased the synthesis of digestive enzymes, improving nutrient digestibility and growth efficiency. Easssawy et al. [36] investigated the effect of dietary supplementation of QSE as a sole source of natural antioxidants in the diet of broilers. The results revealed that the dietary supplemented group with QSE had shown significantly higher body weight, weight gain ratio, and feed consumption as compared to the control. Dietary inclusion of quinoa seed extracts in broiler diets as a natural antioxidant had a good influence on broiler performance, meat quality, and also increased the oxidative stability of chicken meat during refrigerated storage for a longer time.

Sarikaya et al. [37] stated that breast and back-neck ratios in quails were similar in the groups, but there were differences in terms of wing and thigh ratios. In our study, the results obtained in terms of wing ratios were similar to the results reported by Partovi et al. [38].

Nokandi et al. [39] investigated the effect of dietary supplementation of washed and peeled quinoa seed on the performance, nutrient digestibility, and gut morphological response of broilers. The authors reported that the addition of 4% peeled quinoa influenced the rise of body weight, including the breast and thigh, while decreasing abdomen fat. Moreover, the addition of 4% peeled quinoa increased the feed consumption considerably, and the FCR was lower compared to the control group. On the other hand, Tao et al. [40] reported that birds fed a meal supplemented with 12% QSE had significantly lower BWG and unaltered FCR.

## 5. Conclusions

The addition of QSE to the mixture for quails increased feed intake. The carcass yield, thigh, breast, back and neck ratio, and internal organ ratios were not affected by the supplementation. The pH value decreased with an increase in the amount of quinoa seed extract in quail mixed feed. As with the pH values in the same groups, the thiobarbituric acid value and the peroxide value, which is the primary oxidation product, were found to be lower than control in the doped groups on all days of the analysis. These results showed that the antioxidant-effective phenolic substances that quinoa seed extracts possess are effective in preventing or delaying the oxidation of meat and can be used in mixed feed as a natural antioxidant.

According to the presented research, the best results of quail performance were obtained with 0.2 g/kg and 0.4 g QSE/kg of the quail’s fodder. Moreover, the addition of 0.4 g QSE/kg of the quail’s fodder had a significant effect on meat shelf life and could be used in poultry mixed feed to prevent or delay lipid oxidation of meat.

## Figures and Tables

**Table 1 animals-12-01851-t001:** Weekly live weight values of experimental groups (g).

Weeks		Groups				SEM	*p*
		Control	0.1 g/kg QSE	0.2 g/kg QSE	0.4 g/kg QSE		
Per trial		8.44 ± 0.06	8.46 ± 0.07	8.42 ± 0.07	8.43 ± 0.06	0.032	0.980
1		38.75 ± 0.56 ^C^	39.93 ± 0.51 ^BC^	41.85 ± 0.55 ^A^	41.25 ± 0.44 ^AB^	0.268	0.000
2		97.95 ± 1.01	98.35 ± 1.08	99.96 ± 1.07	100.25 ± 0.99	0.521	0.304
3	M	151.64 ± 2.01	154.86 ± 1.85	153.68 ± 1.52	155.62 ± 1.67	0.902	0.386
	F	155.96 ± 2.41	155.29 ± 1.88	158.02 ± 1.92	157.38 ± 2.53	1.064	0.771
	AVG	153.75 ± 1.60	155.13 ± 1.35	156.21 ± 1.31	156.32 ± 1.37	0.705	0.542
4	M	223.80 ± 2.64	227.44 ± 2.32	224.95 ± 3.04	230.44 ± 2.30	1.287	0.235
	F	237.48 ± 3.07	229.06 ± 2.70	235.63 ± 2.79	233.40 ± 3.44	1.494	0.205
	AVG	230.41 ± 2.19	228.33 ± 1.80	231.36 ± 2.16	231.69 ± 1.96	1.014	0.646
5	M	264.03 ± 3.04 ^b^	270.71 ± 3.54 ^ab^	275.48 ± 4.14 ^a^	275.35 ± 2.78 ^a^	1.696	0.050
	F	291.44 ± 6.00 ^b^	293.75 ± 3.32 ^b^	308.75 ± 5.06 ^a^	297.77 ± 4.01 ^ab^	2.453	0.042
	AVG	277.28 ± 3.28 ^B^	283.00 ± 2.83 ^B^	293.78 ± 3.96 ^A^	283.71 ± 2.69 ^B^	1.709	0.006

M: Male; F: Female; AVG: Average; SEM: Standard Mean Error; QSE: quinoa seed extract; ^A, B, C^—highly significant differences at the level of *p* < 0.01; ^a, b^—significant differences at the level of *p* < 0.05.

**Table 2 animals-12-01851-t002:** Weekly live weight gain values of experimental groups (g).

Weeks	Groups				SEM	*p*
	Control	0.1 g/kg QSE	0.2 g/kg QSE	0.4 g/kg QSE		
1	30.31 ± 0.28 ^C^	31.47 ± 0.55 ^BC^	33.42 ± 0.46 ^A^	32.62 ± 0.51 ^AB^	0.367	0.002
2	59.19 ± 1.70	58.41 ± 0.47	58.11 ± 0.52	59.18 ± 1.18	0.503	0.857
3	55.80 ± 0.56	56.78 ± 1.01	56.24 ± 1.94	56.19 ± 2.63	0.780	0.983
4	76.64 ± 0.78	73.20 ± 1.83	76.40 ± 2.96	75.29 ± 2.00	0.978	0.633
5	46.89 ± 3.34	54.66 ± 1.81	60.91 ± 7.32	52.04 ± 0.99	2.321	0.172

SEM: Standard Mean Error; QSE: quinoa seed extract; ^A, B, C^—highly significant differences at the level of *p* < 0.01.

**Table 3 animals-12-01851-t003:** Body weight gains of experimental groups (g).

Groups	Weeks		
	0–2 Week	3–5 Week	0–5 Week
Control	89.50 ± 1.89	179.33 ± 3.39	268.84 ± 3.47
0.1 g/kg QSE	89.88 ± 0.78	184.65 ± 1.88	274.53 ± 2.48
0.2 g/kg QSE	91.54 ± 0.56	193.81 ± 6.76	285.35 ± 6.30
0.4 g/kg QSE	91.80 ± 0.95	183.52 ± 3.02	275.33 ± 3.52
SEM	0.579	2.314	2.422
*p*	0.428	0.150	0.094

SEM: Standard Mean Error; QSE: quinoa seed extract.

**Table 4 animals-12-01851-t004:** Weekly feed consumption of experimental groups (g).

Groups	Weeks				
	1	2	3	4	5
control	56.74 ± 1.52	131.76 ± 1.10 ^B^	134.90 ± 1.10	200.00 ± 0.00	238.63 ± 2.67 ^b^
0.1 g/kg QSE	57.38 ± 0.90	143.32 ± 1.48 ^A^	139.28 ± 1.61	200.00 ± 0.00	246.80 ± 2.66 ^ab^
0.2 g/kg QSE	57.47 ± 2.91	145.48 ± 1.56 ^A^	137.05 ± 1.36	200.00 ± 0.00	248.35 ± 2.60 ^ab^
0.4 g/kg QSE	57.26 ± 0.66	142.04 ± 2.08 ^A^	143.48 ± 4.50	203.57 ± 3.57	254.52 ± 4.36 ^a^
SEM	0.779	1.537	1.402	0.892	2.038
*p*	0.991	0.000	0.157	0.426	0.029

SEM: Standard Mean Error; QSE: quinoa seed extract; ^A, B^—highly significant differences at the level of *p* < 0.01; ^a, b^—significant differences at the level of *p* < 0.05.

**Table 5 animals-12-01851-t005:** Feed consumption of experimental groups (g).

Groups	Weeks		
	0–2 Week	3–5 Week	0–5 Week
Control	188.51 ± 2.05 ^B^	573.53 ± 3.39	762.04 ± 3.34 ^B^
0.1 g/kg QSE	200.71 ± 1.61 ^A^	586.08 ± 1.93	786.80 ± 2.66 ^A^
0.2 g/kg QSE	202.95 ± 1.36 ^A^	585.40 ± 3.74	788.35 ± 2.66 ^A^
0.4 g/kg QSE	199.31 ± 1.99 ^A^	601.57 ± 10.88	800.89 ± 10.10 ^A^
SEM	1.640	3.890	4.425
*p*	0.000	0.067	0.003

SEM: Standard Mean Error; QSE: quinoa seed extract; ^A, B^—highly significant differences at the level of *p* < 0.01.

**Table 6 animals-12-01851-t006:** Carcass values of experimental groups (%).

Groups		Live Weight (g)	Hot Carcass Weight (g)	Cold Carcass Weight (g)	Carcass Yield (%)
Control	Mean	280.18 ± 4.36	209.58 ± 2.08 ^B^	210.22 ± 2.23 ^b^	75.15 ± 0.70
0.1 g/kg QSE		282.06 ± 3.45	212.22 ± 2.44 ^B^	209.28 ± 2.73 ^b^	74.23 ± 0.69
0.2 g/kg QSE		291.37 ± 5.06	222.37 ± 3.18 ^A^	218.18 ± 3.12 ^a^	74.97 ± 0.65
0.4 g/kg QSE		288.87 ± 3.28	222.03 ± 2.35 ^A^	216.45 ± 2.28 ^ab^	74.94 ± 0.31
SEM		2.084	1.436	1.365	0.300
*p*		0.176	0.001	0.043	0.725
Control	M	265.75 ± 2.29 ^b^	203.96 ± 1.44 ^b^	204.03 ± 1.38	76.81 ± 0.73
0.1 g/kg QSE	M	271.00 ± 3.51 ^ab^	209.41 ± 3.24 ^ab^	205.03 ± 3.33	75.64 ± 0.53
0.2 g/kg QSE	M	275.50 ± 3.55 ^ab^	214.72 ± 3.29 ^a^	210.42 ± 2.89	76.37 ± 0.25
0.4 g/kg QSE	M	280.37 ± 3.45 ^a^	215.77 ± 2.55 ^a^	210.88 ± 2.39	75.22 ± 0.33
SEM		1.821	1.550	1.354	0.259
*p*		0.023	0.018	0.157	0.127
Control	F	294.62 ± 4.08	215.20 ± 2.73 ^B^	216.41 ± 2.91	73.49 ± 0.88
0.1 g/kg QSE	F	293.12 ± 1.93	215.03 ± 3.58 ^B^	213.52 ± 3.97	72.82 ± 1.10
0.2 g/kg QSE	F	307.25 ± 5.01	230.02 ± 3.96 ^A^	225.93 ± 4.03	73.57 ± 1.09
0.4 g/kg QSE	F	297.37 ± 3.67	228.28 ± 2.43 ^A^	222.01 ± 2.78	74.66 ± 0.53
SEM		2.075	1.991	1.863	0.457
*p*		0.062	0.002	0.074	0.575

SEM: Standard Mean Error; QSE: quinoa seed extract; A: Mean, F: Female; M: Male; ^A, B^—highly significant differences at the level of *p* < 0.01; ^a, b^—significant differences at the level of *p* < 0.05.

**Table 7 animals-12-01851-t007:** Ratios of main carcass parts in carcass (%).

Groups		Thighs (%)	Breast (%)	Wings (%)	Back (%)	Neck (%)
Control	Mean	32.85 ± 0.30	34.46 ± 0.51	9.15 ± 0.60	14.15 ± 0.32	7.52 ± 0.37
0.1 g/kg QSE		33.13 ± 0.30	33.45 ± 0.39	9.05 ± 0.17	14.80 ± 0.26	7.57 ± 0.22
0.2 g/kg QSE		33.02 ± 0.21	34.00 ± 0.37	9.22 ± 0.12	14.39 ± 035	7.98 ± 0.22
0.4 g/kg QSE		32.90 ± 0.30	34.71 ± 0.53	8.06 ± 0.21	14.38 ± 0.29	7.28 ± 0.25
SEM		0.146	0.230	0.173	0.152	0.121
*p*		0.912	0.228	0.058	0.517	0.237
Control	M	33.35 ± 0.35	33.42 ± 0.65	8.58 ± 0.14 ^bc^	14.45 ± 0.45	7.69 ± 0.43
0.1 g/kg QSE	M	33.28 ± 0.49	33.34 ± 0.53	9.20 ± 0.22 ^ab^	15.28 ± 0.38	7.18 ± 0.20
0.2 g/kg QSE	M	32.96 ± 0.37	33.88 ± 0.60	9.36 ± 0.20 ^a^	14.12 ± 0.28	7.73 ± 0.28
0.4 g/kg QSE	M	33.73 ± 0.44	34.49 ± 0.89	8.39 ± 0.31 ^c^	14.24 ± 0.50	7.35 ± 0.42
SEM		0.202	0.332	0.130	0.212	0.168
*p*		0.630	0.615	0.013	0.206	0.627
Control	F	32.34 ± 0.45	35.51 ± 0.64	8.47 ± 0.18 ^A^	13.85 ± 046	7.35 ± 0.36
0.1 g/kg QSE	F	32.97 ± 0.39	33.56 ± 0.61	8.89 ± 0.27 ^A^	14.31 ± 0.29	7.96 ± 0.34
0.2 g/kg QSE	F	33.09 ± 0.23	34.12 ± 0.47	9.08 ± 0.14 ^A^	14.66 ± 0.64	8.23 ± 0.33
0.4 g/kg QSE	F	32.07 ± 0.42	34.93 ± 0.64	7.74 ± 0.26 ^B^	14.52 ± 0.32	7.20 ± 0.31
SEM		0.196	0.311	0.138	0.220	0.176
*p*		0.201	0.120	0.001	0.612	0.123

SEM: Standard Mean Error; QSE: quinoa seed extract; F: Female; M: Male; ^A, B^—highly significant differences at the level of *p* < 0.01; ^a, b, c^—significant differences at the level of *p* < 0.05.

**Table 8 animals-12-01851-t008:** Edible internal organs and abdominal fat ratios.

Groups		Heart (%)	Liver (%)	Gizzard (%)	Abdominal Fat (%)
Control	Mean	1.06 ± 0.06	3.60 ± 0.21	2.52 ± 0.14 ^b^	1.91 ± 0.17
0.1 g/kg QSE		1.22 ± 0.04	3.98 ± 0.19	2.59 ± 0.11 ^b^	1.93 ± 0.17
0.2 g/kg QSE		1.13 ± 0.05	3.94 ± 0.28	2.57 ± 0.09 ^b^	2.06 ± 0.11
0.4 g/kg QSE		1.15 ± 0.06	4.02 ± 0.21	2.99 ± 0.13 ^a^	1.97 ± 0.20
SEM		0.027	0.111	0.063	0.081
*p*		0.247	0.521	0.029	0.930
Control	M	1.10 ± 0.08	3.08 ± 0.15	2.52 ± 0.15 ^ab^	2.27 ± 0.22
0.1 g/kg QSE	M	1.24 ± 0.03	3.60 ± 0.12	2.34 ± 0.13 ^b^	1.67 ± 0.16
0.2 g/kg QSE	M	1.09 ± 0.09	3.30 ± 0.15	2.45 ± 0.10 ^b^	1.90 ± 0.11
0.4 g/kg QSE	M	1.11 ± 0.12	3.66 ± 0.31	2.87 ± 0.15 ^a^	1.98 ± 0.29
SEM		0.042	0.103	0.073	0.106
*p*		0.574	0.168	0.050	0.261
Control	F	1.02 ± 0.10	4.11 ± 0.31	2.52 ± 0.25	1.56 ± 0.20
0.1 g/kg QSE	F	1.20 ± 0.07	4.36 ± 0.31	2.82 ± 0.14	2.20 ± 0.27
0.2 g/kg QSE	F	1.18 ± 0.06	4.57 ± 0.43	2.70 ± 0.14	2.22 ± 0.19
0.4 g/kg QSE	F	1.20 ± 0.05	4.38 ± 0.23	3.11 ± 0.22	1.97 ± 0.31
SEM		0.037	0.157	0.100	0.125
*p*		0.240	0.795	0.202	0.223

SEM: Standard Mean Error; QSE: quinoa seed extract; F: Female; M: Male; ^a, b^—significant differences at the level of *p* < 0.05.

**Table 9 animals-12-01851-t009:** Peroxide value of breast meat (meq/kg).

Groups	Days			
	1	3	5	7
Control	3.50 ± 0.50 ^A^	6.00 ± 0.00 ^A^	7.50 ± 0.50 ^A^	8.50 ± 0.50 ^A^
0.1 g/kg QSE	3.00 ± 0.00 ^A^	4.00 ± 0.00 ^B^	5.00 ± 0.00 ^B^	6.00 ± 0.00 ^B^
0.2 g/kg QSE	2.00 ± 0.00 ^B^	3.50 ± 0.00 ^B^	4.00 ± 0.00 ^C^	5.00 ± 0.00 ^C^
0.4 g/kg QSE	1.00 ± 0.00 ^C^	2.00 ± 0.00 ^C^	3.00 ± 0.00 ^D^	3.00 ± 0.00 ^D^
SEM	0.375	0.548	0.639	0.754
*p*	0.007	0.002	0.001	0.000

SEM: Standard Mean Error; QSE: quinoa seed extract; ^A, B, C, D^—highly significant differences at the level of *p* < 0.01.

**Table 10 animals-12-01851-t010:** Thiobarbituric acid (TBA) value of breast meat (mg MDA/kg).

Groups	Days				
	0	1	3	5	7
Control	0.04 ± 0.01	0.08 ± 0.00 ^A^	0.27 ± 0.01 ^A^	0.33 ± 0.01 ^A^	0.43 ± 0.02 ^A^
0.1 g/kg QSE	0.04 ± 0.01	0.06 ± 0.00 ^B^	0.19 ± 0.01 ^B^	0.21 ± 0.00 ^B^	0.29 ± 0.01 ^B^
0.2 g/kg QSE	0.03 ± 0.01	0.04 ± 0.00 ^C^	0.12 ± 0.00 ^C^	0.13 ± 0.01 ^C^	0.21 ± 0.02 ^C^
0.4 g/kg QSE	0.02 ± 0.00	0.02 ±0.00 ^D^	0.06 ±0.01 ^D^	0.03 ± 0.01 ^D^	0.15 ± 0.00 ^D^
SEM	0.004	0.006	0.023	0.032	0.032
*p*	0.250	0.000	0.000	0.000	0.000

SEM: Standard Mean Error; QSE: quinoa seed extract; ^A, B, C, D^—highly significant differences at the level of *p* < 0.01.

**Table 11 animals-12-01851-t011:** Total number of psychrophilic bacteria in breast meat (log CFU/g).

Groups	Days				
	0	1	3	5	7
Control	2.22 ± 0.04 ^A^	2.14 ± 0.01 ^A^	3.11 ± 0.38	3.42 ± 0.06 ^A^	3.64 ± 0.04 ^A^
0.1 g/kg QSE	2.07 ± 0.01 ^B^	2.10 ± 0.02 ^B^	3.12 ± 0.14	3.32 ± 0.02 ^A^	3.55 ± 0.03 ^B^
0.2 g/kg QSE	2.06 ± 0.03 ^B^	2.05 ± 0.04 ^B^	2.93 ± 0.03	3.04 ± 0.05 ^B^	3.41 ± 0.03 ^C^
0.4 g/kg QSE	1.65 ± 0.03 ^C^	1.86 ± 0.01 ^C^	2.93 ± 0.03	2.94 ± 0.06 ^B^	3.35 ± 0.03 ^C^
SEM	0.080	0.041	0.083	0.077	0.199
*p*	0.000	0.002	0.840	0.005	0.000

SEM: Standard Mean Error; QSE: quinoa seed extract; ^A, B, C^—highly significant differences at the level of *p* < 0.01.

**Table 12 animals-12-01851-t012:** Effect of storage on breast pH values.

Groups	Days				
	0	1	3	5	7
Control	5.71 ± 0.02 ^A^	5.77 ± 0.00 ^A^	5.98 ± 0.00 ^A^	6.22 ± 0.30 ^a^	6.37 ± 0.04 ^A^
0.1 g/kg QSE	5.64 ± 0.02 ^B^	5.58 ± 0.02 ^B^	5.70 ± 0.01 ^B^	5.65 ± 0.06 ^b^	5.93 ± 0.02 ^B^
0.2 g/kg QSE	5.53 ± 0.02 ^C^	5.48 ± 0.04 ^C^	5.62 ± 0.06 ^B^	5.75 ± 0.02 ^ab^	5.90 ± 0.00 ^B^
0.4 g/kg QSE	5.47 ± 0.01 ^D^	5.38 ± 0.02 ^D^	5.67 ± 0.03 ^B^	5.45 ± 0.02 ^b^	5.85 ± 0.02 ^B^
SEM	0.028	0.044	0.044	0.108	0.004
*p*	0.000	0.000	0.000	0.043	0.000

SEM: Standard Mean Error; QSE: quinoa seed extract; ^A, B, C, D^—highly significant differences at the level of *p* < 0.01; ^a, b^—significant differences at the level of *p* < 0.05.

## Data Availability

The data presented in this study are available on request from the corresponding author.

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
