# Peer review of "The Effect of Quinoa Seed (*Chenopodium quinoa* Willd.) Extract on the Performance, Carcass Characteristics, and Meat Quality in Japanese Quails (*Coturnix coturnix japonica*)"

_animals, 2022, doi:10.3390/ani12141851_

Round 1
Reviewer 1 Report
General comments
The entire document lacked line numbers. Therefore, the individual lines were noted where revisions were requested. The abstract is inaccurate, as it doesn't state what was presented in the paper. In the results there were significant changes in the weight gain and growth of animals on specific treatments. The abstract only stated that there was a significant difference in feed intake. The abstract should be re-written to express the real information of the document. In the introduction nothing was mentioned about the quail. Why was this bird used? Does this bird have any nutritional benefits to human nutrition without the supplementation of QSE? The were many grammatical and English errors present throughout the document. Some have been highlighted in the specific comments (too numerous to highlight all; also present in the results and discussion section but not highlighted) but I believe that the paper does require English editing from a reputable source before it can be further processed. In the statistical analysis, if weight (and other parameters) of the birds are being measured over time then analysis of variance using repeated measures should be used. This will give information on how the treatments also interact with time. Thus analysis of this data should also be re-done.
After considering the many faults in the manuscript. The topic is quite novel and if the extensive revisions are made it can be published.
Specific comments
Simple summary:
Please revise : "Quinoa seed is a good source of vital amino acids like lysine and methionine, as well as have a high protein quality."
How can it be said that QSE has an good effect on the growth of the birds when this wasn't statistically significant?
Abstract:
Should be re-written. Providing accurate information.
Introduction:
Please revise: "Although synthetic antioxidants and antimicrobials have been approved in many countries to prevent undesirable reactions and extend the shelf life of the product". The word 'although' can be omitted.
Please revise: "Phytobiotics were used to promote the growth and improve the carcass quality with less fat content in the broiler meat"
"Antibiotic remains found in meat and eggs may cause complications with human health" Can this be expanded a bit giving specific examples?
The following can be separated into two sentences;
" Oxidation of meat lipids decreasing the shelf life of the meat, deteriorating the taste and standards of the food and also affecting the organoleptic properties are the biggest problems encountered in meat processing, cooking and cold storage."
It can be revised to, "Oxidation of meat lipids decreasing the shelf life of the meat, deteriorating the taste and standards of the food and also affecting the organoleptic properties. Which are the biggest problems encountered in meat processing, cooking and cold storage"
Please revise: To ensure high quality food products, it should be continued by adding antioxidants (natural or synthetic) that do not allow oxidation of lipids [9, 10].
Please revise: After the prohibition of using antibiotics as a growth promoter in the poultry sector, the search for natural and safe substances that do not have any negative effects on human and animal health. What information is this sentence trying to provide?
Methods (2.3):
Na2CO3 solution (200 mg mL) should read 200mg/mL. Also, check throughout manuscript that units are correct.
Please revise: "Add 10 mL of ultra-pure water, Then sediment was separated by centrifugation for 5 minutes at 4000 rpm" The authors should remember that scientific writing is written in third person, past tense, singular or plural in a passive voice.
Please revise: "A 10 µl of sample was added into 1 ml of ABTS+•"
Please revise: "In order to determine the weekly live weight gains of quails, it was calculated by subtracting the average live weight of the previous week from the average live weight of each replication in each group determined in the weighting for each week." This sentence is not clear and there are spelling errors.
Method (2.6)
Is this correct? "This oxidation analysis included three parameters: peroxide value analysis and thiobarbituric acid number (TBARS)." You stated three parameters, however, only two are mentioned.
Please revise: "for1 min"
Should the following paragraph be placed in microbial analysis?
"The total psychrophilic counts of the quail breast flesh samples were conducted at 0, 3, 5, and 7 days. For this, 10 g of the quail breast flesh samples was homogenized for1 min in 90 mL of 0.1 percent peptone water. Diluting the homogeneous mixture with 0.1 percent peptone water and percent ringer solution yielded serial dilutions. In the total psychrophilic live count study, the plate count agar (PCA) medium was employed. For 7 days, Petri dishes were incubated at 7℃. Log CFU/g is the unit of measurement for the number of microbiological bacteria."
Method (2.8)
Please revise: "At the end of the research, 3 breast meat was taken from each group of quails"
Please revise: "At the end of the day storage periods, the pH values on all days"
Please revise: "thepH value"
Results (3.2)
Please revise: "the average body weight in each group was tried to be similar"
Author Response
Replies to the reviewer - animals-1809832
Dear Reviewer "Please see the attachment."
To whom it may concern,
The authors would like to thank the reviewers for all comments and suggestions which may improve manuscript quality. The authors made changes to the text in line with the reviewers' renewed comments.
Reviewer 1
General comments
"The entire document lacked line numbers. Therefore, the individual lines were noted where revisions were requested".
We have added line numbers to the entire document. The authors would like to say thank you to reviewer 1 for highlighting this important point.
The abstract is inaccurate, as it doesn't state what was presented in the paper. In the results there were significant changes in the weight gain and growth of animals on specific treatments. The abstract only stated that there was a significant difference in feed intake. The abstract should be re-written to express the real information of the document.
We have revised the abstract according to reviewer 1 valuable suggestions to improve the quality of the manuscript. The authors would like to thank you for catching this point.
In the introduction nothing was mentioned about the quail. Why was this bird used? Does this bird have any nutritional benefits to human nutrition without the supplementation of QSE?
We have added information about quail….Quails inhabit regions of Asia, America, Europe, and Australia, however remunerative quail breeds are raised for eggs and meat intention around the globe. Japanese quails (Coturnix coturnix japonica) have the ability to attain a live weight of 200 grams within 4 weeks of age. In a free-range rearing system, the weight of this bird is around 100-160 grams [1]. Regardless of the quality, quail farming is very expedient due to the least cost of maintenance, healthy production, and a remarkable revenue ratio [2]. The different quail products have shown numerous pro-health properties, helpful in the treatment of different diseases like ulcers or gastritis, consolidation of heart muscles, and rehabilitation of blood circulation after blood stroke, and had antineoplastic effects.
The were many grammatical and English errors present throughout the document. Some have been highlighted in the specific comments (too numerous to highlight all; also present in the results and discussion section but not highlighted) but I believe that the paper does require English editing from a reputable source before it can be further processed.
Every effort has been made to improve the quality of the language and the legibility of the text.
Specific comments
Simple summary:
Please revise : "Quinoa seed is a good source of vital amino acids like lysine and methionine, as well as have a high protein quality." How can it be said that QSE has an good effect on the growth of the birds when this wasn't statistically significant?
We have revised this valuable suggestion of the reviewer. The authors would like to thank the reviewer to highlight this point.
Abstract:
Should be re-written. Providing accurate information.
Following the reviewer's suggestions, information has been added, which should improve the readability of the abstract. We have revised the abstract according to the suggestions of the reviewer. The authors hope that thanks to this, the abstract of the manuscript is easy to read and understand for the reader.
Introduction:
Please revise: "Although synthetic antioxidants and antimicrobials have been approved in many countries to prevent undesirable reactions and extend the shelf life of the product". The word 'although' can be omitted.
The authors have revised this line. The authors would like to thank you for highlighting this point.
Please revise: "Phytobiotics were used to promote the growth and improve the carcass quality with less fat content in the broiler meat"
We have revised this line as well according to reviewer 1 valuable suggestion.
"Antibiotic remains found in meat and eggs may cause complications with human health" Can this be expanded a bit giving specific examples?
Yes, we have added some information for this point as well and revised this line also. Thank you for the reviewer to highlight this valuable information.
The following can be separated into two sentences;
" Oxidation of meat lipids decreasing the shelf life of the meat, deteriorating the taste and standards of the food and also affecting the organoleptic properties are the biggest problems encountered in meat processing, cooking and cold storage."
We have separated this into 2 lines according to 1 reviewer's suggestion.
It can be revised to, "Oxidation of meat lipids decreasing the shelf life of the meat, deteriorating the taste and standards of the food and also affecting the organoleptic properties. Which are the biggest problems encountered in meat processing, cooking and cold storage"
We have revised this line according to reviewer 1 valuable suggestion.
Please revise: To ensure high quality food products, it should be continued by adding antioxidants (natural or synthetic) that do not allow oxidation of lipids [9, 10].
We have revised this line, to improve the quality of the manuscript according to reviewer 1 valuable suggestion.
Please revise: After the prohibition of using antibiotics as a growth promoter in the poultry sector, the search for natural and safe substances that do not have any negative effects on human and animal health. What information is this sentence trying to provide?
We have revised this whole sentence and added some more information related to this.
Methods (2.3):
Na2CO3 solution (200 mg mL) should read 200mg/mL. Also, check throughout manuscript that units are correct.
The authors have checked all the units in the entire manuscript now and corrected this line as well. The authors would like to appreciate the suggestions of reviewer 1.
Please revise: "Add 10 mL of ultra-pure water, Then sediment was separated by centrifugation for 5 minutes at 4000 rpm" The authors should remember that scientific writing is written in third person, past tense, singular or plural in a passive voice.
The authors have revised this line. Thanks for your valuable suggestions.
Please revise: "A 10 µl of sample was added into 1 ml of ABTS+•"
The authors have revised this line. Thanks for the valuable suggestions of reviewer 1.
Please revise: "In order to determine the weekly live weight gains of quails, it was calculated by subtracting the average live weight of the previous week from the average live weight of each replication in each group determined in the weighting for each week." This sentence is not clear and there are spelling errors.
Method (2.6)
Is this correct? "This oxidation analysis included three parameters: peroxide value analysis and thiobarbituric acid number (TBARS)." You stated three parameters, however, only two are mentioned.
Thanks for highlighting our mistake. We have revised this line and correct this according to instructions of reviewer 1.
Please revise: "for1 min"
We have revised this line.
Should the following paragraph be placed in microbial analysis?
"The total psychrophilic counts of the quail breast flesh samples were conducted at 0, 3, 5, and 7 days. For this, 10 g of the quail breast flesh samples was homogenized for1 min in 90 mL of 0.1 percent peptone water. Diluting the homogeneous mixture with 0.1 percent peptone water and percent ringer solution yielded serial dilutions. In the total psychrophilic live count study, the plate count agar (PCA) medium was employed. For 7 days, Petri dishes were incubated at 7℃. Log CFU/g is the unit of measurement for the number of microbiological bacteria."
We have shifted this paragraph to the microbial analysis section. The authors would like to say thanks for highlighting this concern to improve the quality of the manuscript.
Method (2.8)
Please revise: "At the end of the research, 3 breast meat was taken from each group of quails"
The authors have revised this line.
Please revise: "At the end of the day storage periods, the pH values on all days"
The authors have revised this line.
Please revise: "thepH value"
The authors have this line.
Results (3.2)
Please revise: "the average body weight in each group was tried to be similar"
The authors have revised this line. The authors would like to say thank you for all the valuable comments and suggestions to improve the quality of our manuscript.
The authors hope that the corrections made in line with the reviewers' suggestions have improved the readability and quality of the manuscript.
Yours faithfully,
Authors

Reviewer 2 Report
The paper is very well written and includes all the elements of an original scientific paper. The research design is appropriate, and the results were clearly presented. The only objection is that in the conclusion, and subsequently not even in the summary, the author's recommendation on the use of the QSE portion in quail feeding according to the presented results is not given. Therefore, I recommend that the author should supplement the Conclusion and Summary with a strict recommendation on the using of the QSE (amount in the feed mixture) in the feeding of quails.
Author Response
Replies to the reviewer - animals-1809832
Dear Reviewer "Please see the attachment."
To whom it may concern,
The authors would like to thank the reviewers for all comments and suggestions which may improve manuscript quality. The authors made changes to the text in line with the reviewers' renewed comments.
Reviewer 2
Comments and Suggestions for Authors
The paper is very well written and includes all the elements of an original scientific paper. The research design is appropriate, and the results were clearly presented. The only objection is that in the conclusion, and subsequently not even in the summary, the author's recommendation on the use of the QSE portion in quail feeding according to the presented results is not given. Therefore, I recommend that the author should supplement the Conclusion and Summary with a strict recommendation on the using of the QSE (amount in the feed mixture) in the feeding of quails.
Following the reviewer's suggestions, information has been added, which should improve the readability of the summary, abstract, and conclusion sections. The authors have revised this line. The authors strongly recommended the concentrations of the use of the QSE portion in quail feeding according to the presented results. The authors would like to say thank you for all the valuable comments and suggestions to improve the quality of our manuscript.
The authors hope that the corrections made in line with the reviewers' suggestions have improved the readability and quality of the manuscript.
Yours faithfully,
Authors

Round 2
Reviewer 1 Report
The authors have addressed most of the of the issues present in the paper..